# Giant transverse magnetic fluctuations at the edge of re-entrant superconductivity in UTe$_2$

Valeska Zambra ®[1], Amit Nathwani ®[1,2], Muhammad Nauman ®[1,3], Sylvia K. Lewin ®[4,5], Corey E. Frank ®[4,5], Nicholas P. Butch ®[4,5], Arkady Shekhter[6], B. J. Ramshaw ®[7,8] & K. A. Modic ®[1] ✉

UTe$_2$ exhibits the remarkable phenomenon of re-entrant superconductivity, whereby the zero-resistance state reappears above 40 tesla after being suppressed with a field of around 10 tesla. One potential pairing mechanism, invoked in the related re-entrant superconductors UCoGe and URhGe, involves transverse fluctuations of a ferromagnetic order parameter. However, the requisite ferromagnetic order—present in both UCoGe and URhGe—is absent in UTe$_2$, and neutron scattering shows instead that the magnetic susceptibility is peaked at an antiferromagnetic wavevector. Here, we measure the magnetotropic susceptibility of UTe$_2$ across two field-angle planes. This quantity is sensitive to the magnetic susceptibility in a direction transverse to the applied magnetic field—a quantity that is not accessed in conventional magnetization measurements. We observe a very large decrease in the magnetotropic susceptibility over a broad range of field orientations, indicating a large increase in the transverse magnetic susceptibility. Because our technique probes the magnetic susceptibility in the long wavelength ($q = 0$) limit, this suggests that the strong transverse susceptibility arises from ferromagnetic spin fluctuations. These ferromagnetic fluctuations are likely important for understanding the pairing mechanism in UTe$_2$, as all three superconducting phases of UTe$_2$ surround this region of enhanced susceptibility in the field-angle phase diagram.

Understanding the connection between magnetism and superconductivity in UTe$_2$ is key to determining its superconducting order parameters and pairing mechanisms. A compelling feature of UTe$_2$ is the re-emergence of superconductivity at high magnetic fields in a "halo" of field angles around the $b$-axis[1]. This re-emergence is coincident with a metamagnetic transition to a spin-polarized state (see Fig. 1a). A remarkable feature of both the re-entrant and spin-polarized phases is that they occur in samples that are too disordered to exhibit zero-field superconductivity, and the reentrant superconducting

phase appears to have a higher $T_c$[2-4]. This suggests that the pairing mechanism of the re-entrant superconducting phase may be related to the metamagnetic transition.

A comparison can be made with the uranium-based superconductors UCoGe and URhGe, which also exhibit field re-entrant and field-reinforced superconductivity[5]. Unlike UTe$_2$, UCoGe and URhGe host ferromagnetic order that onsets at temperatures above the superconducting $T_c$[6]. In these compounds, it has been suggested that a magnetic field applied perpendicular to the magnetic easy axis induces

[1]Institute of Science and Technology Austria, Klosterneuburg, Austria. [2]California Institute of Technology, Pasadena, CA, USA. [3]Department of Physics and Astronomy, School of Natural Sciences (SNS), National University of Sciences and Technology (NUST), Islamabad, Pakistan. [4]NIST Center for Neutron Research, National Institute of Standards and Technology, Gaithersburg, MD, USA. [5]Department of Physics, Quantum Materials Center, University of Maryland, College Park, MD, USA. [6]Los Alamos National Laboratory, New Mexico, USA. [7]Laboratory of Atomic and Solid State Physics, Cornell University, Ithaca, NY, USA. [8]Canadian Institute for Advanced Research, Toronto, ON, Canada. ✉e-mail: kimberly.modic@ista.ac.at

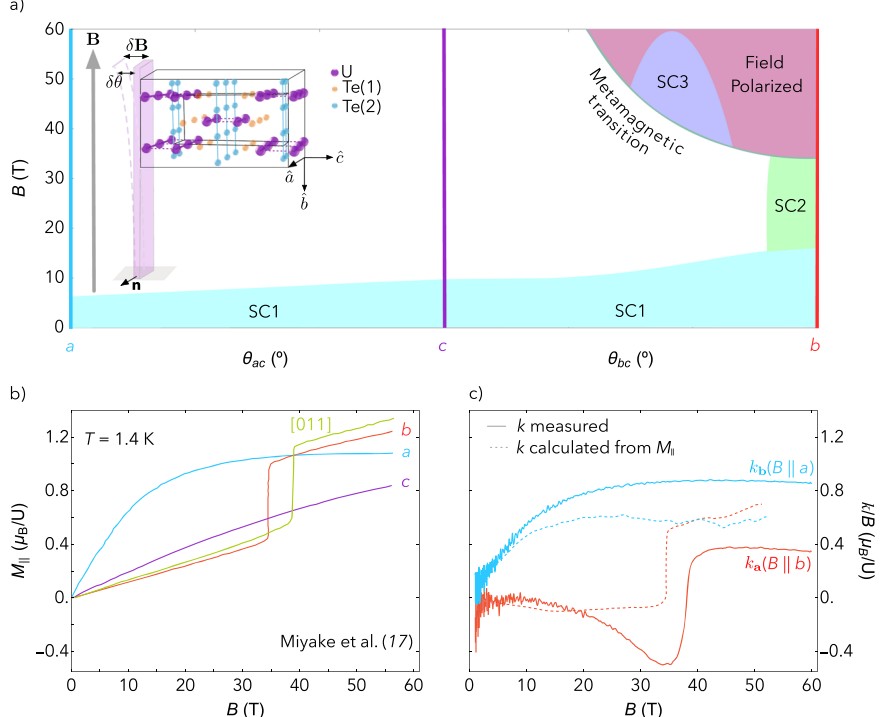

**Fig. 1 | Phase diagram, magnetization, and magnetotropic susceptibility. a** The field-angle phase diagram of UTe$_2$ at 300 mK, showing the field-angle phase boundaries of the SC1, SC2 and SC3 superconducting states (data reproduced from[1,27,28]). The colored vertical lines indicate directions where the data in (**b**) and (**c**) were measured. The inset shows the experimental geometry, where **B** is the applied magnetic field, and the transverse field component $\delta$**B** results from the vibration of the cantilever by the angle $\delta\theta$ around the axis **n**. An example sample orientation is shown for measurements in the $bc$ plane, measuring $k_a$ (**b**) Magnetization as a function of magnetic field applied along each of the crystallographic directions, and in the [011] direction, at $T$ = 1.4 K (data reproduced from[15]). At low field, the $a$-axis is the magnetic easy axis. At the metamagnetic transition $B_m \approx 35$ T, the magnetization for a field along the $b$-axis jumps to -1 $\mu_B$ per uranium and crosses

the $a$-axis magnetization. **c** The magnetotropic susceptibility divided by magnetic field $k/B$, measured at $T$ = 4 K, as a function of field for two different experiment geometries: with the oscillating field component $\delta$**B** in the $ac$-plane measuring $k_b$ (blue curve), and the oscillating field component $\delta$**B** in the $bc$-plane measuring $k_a$ (red curve). The subscripts on $k$ denote the normal vector **n** to the vibration/rotation plane. The dashed lines are calculated using Eq. (1) and the measured magnetization from panel b (e.g., $k_a(\mathbf{B}||\mathbf{b}) = B\left(M_b - B\frac{\partial M_c}{\partial B_c}\right)$). The calculated magnetotropic susceptibility captures the overall qualitative behavior of the measured magnetotropic susceptibility $k_a(\mathbf{B}||\mathbf{b})$ and $k_b(\mathbf{B}||\mathbf{a})$---differences may be attributed to the presence of transverse susceptibility components that are not captured in the magnetization measurements.

transverse fluctuations of the ordered moment that drive spin-triplet pairing[7]. However, UTe$_2$ lacks ferromagnetic order at zero field, and there is no support for fluctuations of the kind relevant to the re-entrant superconductivity found in other uranium-based super-conductors. Instead, neutron scattering experiments find evidence for antiferromagnetic fluctuations in UTe$_2$ at zero and up to at least 11 T[8–10]. While NMR has suggested the presence of longitudinal ferro-magnetic fluctuations from a field-induced phase, these measure-ments have been confined to a field along the $b$-axis[11,12]. In addition, previous theories of field-induced superconductivity focused on transverse fluctuations[7], which have not yet been reported. This calls for further study of the magnetism of UTe$_2$ in high magnetic fields.

Our study focuses on the field-angle phase diagram of UTe$_2$ at 4 kelvin, where all superconducting phases are suppressed and where the magnetic response can be studied in isolation. We measure the magnetotropic susceptibility[13,14], which is sensitive to the magnetic susceptibility in a direction perpendicular to the applied magnetic field. We measure in pulsed magnetic fields up to 60 T, and in a full range of angles spanning both the $ac$- and $bc$-planes. We find evidence for transverse magnetic fluctuations in both planes, with particularly large transverse fluctuations in a region of the $bc$-plane that lies at the edge of all three superconducting phases.

## Results

We first describe the experimental geometry and why the magneto-tropic susceptibility is sensitive to the transverse magnetic

susceptibility. The inset of Fig. 1a shows the experimental geometry for a magnetotropic susceptibility measurement[13,14]. A silicon micro-cantilever is driven near its fundamental bending mode, defining an axis **n** around which the tip of the lever oscillates by a small angle $\delta\theta$. We place the cantilever in an external magnetic field **B** and rotate the cantilever in the plane normal to the vector **n**. This constrains the field angle $\theta$ to always lie in the same plane as the lever oscillation angle $\delta\theta$.

The sample is placed on the tip of the cantilever with one crys-tallographic axis aligned along the length of the lever and another crystallographic axis aligned along **n**. For an orthorhombic crystal like UTe$_2$, this places the third axis perpendicular to the surface of the lever (inset of Fig. 1a). We perform two sets of rotation experi-ments: one in the $ac$-plane and one in the $bc$-plane. In all experiments reported here, the $c$-axis is perpendicular to the surface of the cantilever, and $\theta$ is defined as the angle between the applied magnetic field and the $c$-axis.

When a magnetic field is applied along one of the crystallographic axes of UTe$_2$, it produces a magnetic moment parallel to that axis. When the magnetic field is rotated away from that axis, the moment is no longer parallel to the magnetic field. This produces a magnetic torque, $\tau$ = **M** × **B**, that bends the cantilever by a small angle. The angular derivative of this torque defines the magnetotropic suscept-ibility: $k \equiv \partial\tau/\partial\theta$. This susceptibility adds to the elastic bending stiffness of the cantilever and is measured by the shift in the cantilever reso-nance frequency (see SI for details of the calibration procedure). In terms of the vector **n** around which the cantilever oscillates in an

applied field **B**, the magnetotropic susceptibility is

$$k_\mathbf{n}(\mathbf{B}) = (\mathbf{n} \times \mathbf{B}) \cdot (\mathbf{n} \times \mathbf{M}) - \frac{1}{\mu_0}(\mathbf{n} \times \mathbf{B}) \cdot \chi(\mathbf{B}) \cdot (\mathbf{n} \times \mathbf{B}), \quad (1)$$

where $\chi_{ij}(\mathbf{B}) \equiv \mu_0 \partial M_i(\mathbf{B})/\partial B_j$ is the differential magnetic susceptibility probed by the oscillating magnetic field component that is perpendicular to the applied field $\mathbf{B}$[14]. This oscillating field component arises due to the reorientation of the sample with respect to the field due to the cantilever vibration. The first term in Eq. (1) captures how the torque changes when a fixed moment **M** is rotated in a field **B** around the axis **n**. The second term captures how the torque changes due to the change in the moment itself as the crystal is rotated in the field.

We can now illustrate why $k$ is sensitive to the transverse magnetic susceptibility. The second term of Eq. (1) selects the susceptibility tensor component that is perpendicular to both **n** and **B**. For field along the $c$-axis, and with the sample oscillating in the $ac$-plane ($bc$-plane), this selects $\chi_{aa}(\mathbf{B}\|\mathbf{c})$ ($\chi_{bb}(\mathbf{B}\|\mathbf{c})$). These are what we define as transverse magnetic susceptibility components. Note that these are not off-diagonal susceptibilities, such as $\chi_{bc}(\mathbf{B}\|\mathbf{c})$, which are not allowed in an orthorhombic crystal structure for field along crystal axes. Instead, $\chi_{aa}(\mathbf{B}\|\mathbf{c})$ and $\chi_{bb}(\mathbf{B}\|\mathbf{c})$ are longitudinal (diagonal) susceptibility components that are measured perpendicular to the static, applied magnetic field. The oscillating field component perpendicular to the external field direction is generated by the oscillation of the cantilever with a period of order 20 microseconds (see $\delta\mathbf{B}$ in the inset of Fig. 1a). Further experimental details are given in the Supplementary Information (SI).

Figure 1c shows the measured magnetotropic susceptibility divided by magnetic field for two different crystal orientations on the cantilever at $T = 4$ K, and with field applied along two different axes: $k_\mathbf{a}(\mathbf{B}\|\mathbf{b})$ and $k_\mathbf{b}(\mathbf{B}\|\mathbf{a})$ (see SI for details of the sample orientations). Figure 1b shows the measured magnetization along each of the principal crystallographic directions for comparison (reproduced from ref. 15). The metamagnetic transition is clearly visible near 35 tesla for $\mathbf{B}\|\mathbf{b}$ in both the magnetization and the magnetotropic susceptibility measurements.

The magnetization in Fig. 1b is the longitudinal magnetization: it is found by integrating the magnetic susceptibility measured along the applied field direction, $M_i = \int (\partial M_i/\partial B_i)\, dB_i$[15,16]. We use the longitudinal magnetization and its field derivative—the longitudinal magnetic susceptibility—in conjunction with Eq. (1) to calculate the expected magnetotropic susceptibility. This procedure does not account for any nonlinear transverse component to the magnetic susceptibility tensor (these components will be important later). This calculation is shown as dashed lines in Fig. 1c for $\mathbf{B}\|\mathbf{a}$ and $\mathbf{B}\|\mathbf{b}$. The overall magnitude and the qualitative features of the calculated and measured magnetotropic susceptibility are in good agreement. Small differences can be attributed to a small misalignment between the rotation vector $n$, the plane of the cantilever, and the sample's crystal axes (for more details regarding alignments, see SI II). This demonstrates that, for field along the $a$- and $b$-axes, the measured magnetotropic susceptibility is largely determined by the longitudinal magnetic susceptibility. As shown below, this will not be the case for other field orientations.

Figure 2a and b show the magnetotropic susceptibility for magnetic field applied along the $c$-axis, for both the $k_\mathbf{a}$ and $k_\mathbf{b}$ configurations. We also reproduce the data and calculations from Fig. 1c for comparison. Unlike the other two field orientations, the measured magnetotropic susceptibility for $\mathbf{B}\|\mathbf{c}$ deviates strongly from the estimate made using the longitudinal magnetization and susceptibility alone (dashed purple line). The large negative response in the magnetotropic susceptibility compared to that inferred from magnetization measurements indicates that a new susceptibility component, hidden from the longitudinal magnetization measurements, becomes

active in the magnetotropic susceptibility at a field scale of around 20 tesla.

Panels c and d in Fig. 2 show the evolution of the magnetotropic susceptibility for a broad range of field angles in the $ac$- and $bc$-planes. The large decrease in the magnetotropic susceptibility that onsets near 20 T for field along the $c$-axis is observed over a broad range of angles in both planes. In the $bc$-plane (Fig. 2d), the decrease in the magnetotropic susceptibility is abruptly truncated by the metamagnetic transition into the field-polarized phase (red shaded region). As magnetic field is rotated away from the $b$-axis, the metamagnetic transition moves to higher field and the magnitude of the jump in $k$ increases before abruptly disappearing around 56°. Because $k$ is a susceptibility (i.e., a second derivative of the free energy with respect to angle), the jump is expected to grow in size as the metamagnetic transition becomes more second-order on approach to a critical endpoint. The largest jump in $k$ appears at the angle where the jump in the magnetization (a first derivative) goes to zero[17]. These observations confirm the critical endpoint of the field-polarized phase as first identified by ref. 17.

## Discussion

We uncover the origin of the large decrease in the magnetotropic susceptibility near 20 tesla by first analyzing the data along a high-symmetry direction. When the magnetic field is applied along a crystal axis, the first term in Eq. (1) is completely determined by the longitudinal magnetization. Here, the second term has contributions only from the transverse magnetic susceptibility, i.e. $\chi_{ii}(\mathbf{B}\|\mathbf{j})$, for $j \neq i$. The longitudinal magnetization, shown in Fig. 1b, clearly shows no features that resemble the strong downturn seen in $k$ near 20 tesla. Therefore, the decrease in $k$ for field applied along the $c$-axis must originate from a transverse component of the magnetic susceptibility, i.e. $\chi_{ii}(\mathbf{B}\|\mathbf{j})$.

To highlight the large magnitude of the transverse magnetic susceptibility, we convert our magnetotropic data to dimensionless susceptibility units by dividing $k$ by $B^2/\mu_0$ (Fig. 3). Next, we subtract out the contributions calculated using the longitudinal magnetization and magnetic susceptibility. The remaining contribution is the transverse magnetic susceptibility. Figure 3 shows this susceptibility for $k_\mathbf{a}$ measured with $\mathbf{B}\|\mathbf{c}$, i.e., $\chi_{bb}(\mathbf{B}\|\mathbf{c})$. We also show the longitudinal susceptibility for the same field orientation, i.e., $\chi_{cc}(\mathbf{B}\|\mathbf{c})$. By 40 tesla, the transverse magnetic susceptibility is more than 30 × larger than the longitudinal susceptibility; quantitatively, the magnetotropic susceptibility is almost entirely dominated by the transverse susceptibility.

Moving away from the $c$-axis, the large increase in transverse susceptibility persists, and even strengthens, as we move towards the $b$-axis (Fig. 4). Like the longitudinal magnetization measured along the principal axes, the longitudinal magnetization measured at these intermediate angles indicates no substantial changes in the longitudinal susceptibility[15]. Therefore, the large decrease in $k$ at all angles must come from an increase in the transverse magnetic susceptibility. While we do not have longitudinal magnetization measurements at all angles, and thus cannot subtract out the longitudinal component at all angles, Fig. 3 demonstrates that the transverse component is overwhelmingly larger than the longitudinal component in our measurement and that the transverse susceptibility is essentially equal to $-\mu_0 k/B^2$. We plot this quantity in Fig. 4 for angles in the $ac$- and $bc$-planes. The transverse susceptibility is large for a broad region of the $bc$-plane, and terminates at the boundary of the field-polarized phase.

Our measurements probe the magnetic susceptibility in the low-frequency, long-wavelength limit, $\chi(\omega, q \approx 0)$, and thus we are primarily sensitive to ferromagnetic fluctuations. Transverse ferromagnetic fluctuations naturally emerge when a ferromagnetic phase is destabilized by a magnetic field applied perpendicular to its ordered moment[7,18]. Although UTe$_2$ is not a ferromagnet at zero field, the metamagnetic transition involves a sharp increase in the magnetic

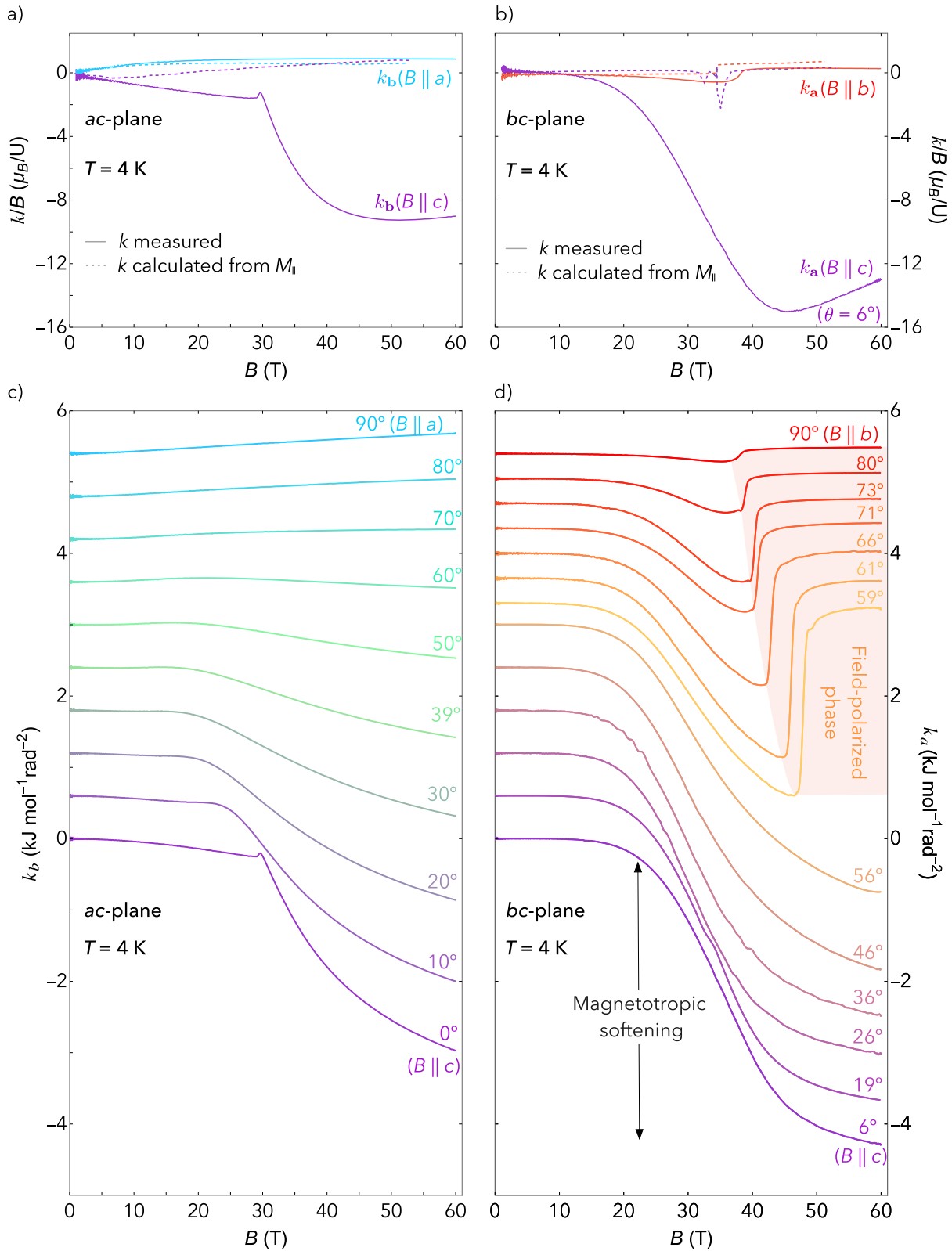

**Fig. 2 | Angle-dependent magnetotropic susceptibility.** Magnetotropic susceptibility measurements performed in pulsed magnetic fields up to 60 T. Panel a) shows $k_b(\mathbf{B}||\mathbf{c})$ and compares it to $k_b(\mathbf{B}||\mathbf{a})$, and likewise for $k_a(\mathbf{B}||\mathbf{c})$ and $k_a(\mathbf{B}||\mathbf{b})$ in panel b). The dashed lines in both panels are the calculated values of $k$ based on the measured longitudinal magnetic susceptibility data from Fig. 1b and Eq. (1) **c** and **d** show the magnetotropic susceptibility measured at multiple angles in the *ac*- and *bc*-planes, respectively. As the magnetic field approaches the *c*-axis in both planes, a large decrease is observed in $k$ that onsets at roughly 20 T. This decrease persists for a range of angles in both planes around **B**||**c**, and is abruptly cut off by the metamagnetic transition into the field-polarized phase (red shaded region in **d**) at $\theta = 59°$ in the *bc* plane.

moment along the *b*-axis. In this sense, the field-polarized phase can be viewed as a "field-induced ferromagnet." Tilting the field away from the *b*-axis introduces a transverse component that ultimately suppresses the first-order transition at a quantum critical endpoint[17]. This behavior strongly echoes that of the canonical ferromagnetic superconductors UCoGe and URhGe, except that in those systems ferromagnetism exists in zero applied field, whereas in UTe$_2$ it appears only under a 35 tesla longitudinal field.

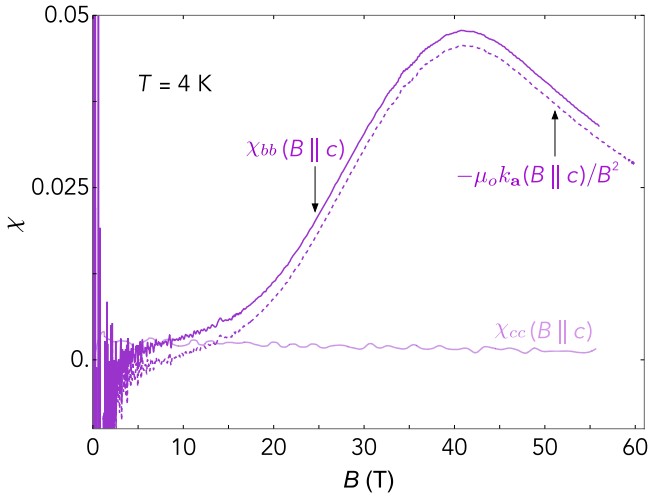

**Fig. 3 | Longitudinal and transverse magnetic susceptibility.** The dimensionless magnetic susceptibilities as a function of magnetic field. The light purple curve shows $\chi_{cc} = \mu_o(\partial M_c/\partial B_c)$ obtained from conventional measurements of the longitudinal magnetic susceptibility, with field applied along the *c*-axis[15]. The dark purple curve shows the transverse magnetic susceptibility calculated from the measured magnetotropic susceptibility $k_a$ and the measured longitudinal magnetic susceptibility $\chi_{cc}$: $\chi_{bb}(\mathbf{B}||\mathbf{c}) = \mu_o(\chi_{cc}(\mathbf{B}||\mathbf{c}) - k_a(\mathbf{B}||\mathbf{c})/B^2)$. Note that the magnetotropic measurements are performed with **B** applied -6° from the *c*-axis, whereas the magnetic susceptibility measurements are performed with **B**||**c**. The dashed purple curve is the measured magnetotropic susceptibility in dimensionless units without the longitudinal magnetic contribution subtracted. The similarity of the solid and dashed curves illustrates how the transverse susceptibility dominates *k* at high magnetic fields.

The presence of strong transverse ferromagnetic fluctuations is significant because such fluctuations are widely believed to promote spin-triplet pairing[7]. This pairing mechanism was suggested to explain the high-field superconductivity in UCoGe and URhGe[6,16], which, as noted above, are easy-axis ferromagnets that exhibit field-reentrant superconductivity when a magnetic field is applied transverse to the ordered moment. We suggest that similar physics may be at play in UTe$_2$−transverse fluctuations in the vicinity of $B^\star$ may provide the "glue" for pairing in the high-field superconducting state. The same fluctuations could also be responsible for the increase in the $T^2$-coefficient of resistivity near the metamagnetic transition[19], as well as the *T*-linear resistivity near SC3 at temperatures above $T_c$[20].

Interestingly, the maximum in $\chi_\perp$ does not coincide exactly with $B^\star$. Several factors may contribute: our measurements were performed at 4 K−well above $T_c$−and probe only one direction in the plane transverse to the applied magnetic field. The true maximum could occur at $B^\star$ for a different transverse orientation. Moreover, fluctuations are only part of the pairing story; the electronic structure also matters[21,22]. For instance, a Zeeman-driven van Hove singularity for one spin species could enhance the density of states elsewhere in the field-angle phase diagram[23].

Magnetotropic susceptibility measurements provide a new window into the high-field phase diagrams of exotic superconductors. Unlike conventional magnetization measurements, they capture the transverse response, and they are compatible with pulsed fields and rotation studies. Future work will map additional field-angle planes in UTe$_2$ while varying the **n** vector orientation to fully resolve the transverse susceptibility tensor. These experiments will clarify how ferromagnetic fluctuations interact with the electronic structure and ultimately drive high-field superconductivity in UTe$_2$.

## Methods
### Crystal synthesis
Crystals were synthesized using the chemical vapor transport method, with iodine as the transport agent and a 5:9 starting ratio of U:Te. The sample was grown in a temperature gradient of 900/830 °C for two weeks. Between measurements, the samples were stored in a vacuum. Further details of the synthesis method are given in ref. 24.

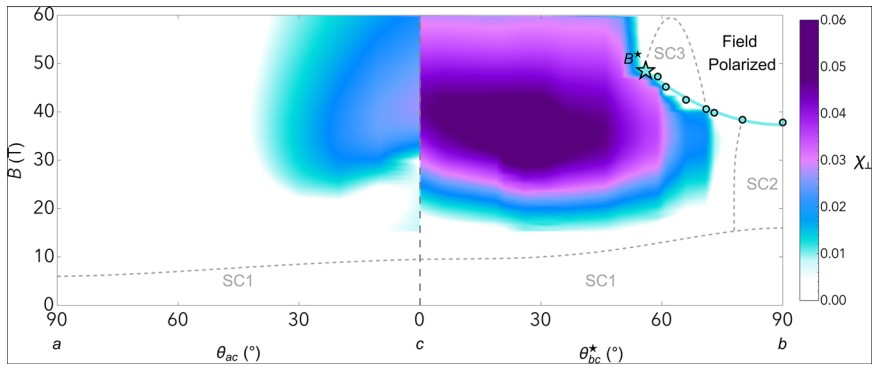

**Fig. 4 | Phase diagram of the magnetotropic susceptibility, $k_b$ in the *ac*-plane and $k_a$ in the *bc*-plane, divided by $B^2$.** A color plot of the magnitude of the magnetotropic susceptibility converted to dimensionless susceptibility units as a function of field strength and field angle in the *ac*- and *bc*-plane at 4 kelvin. The measured magnetotropic susceptibility *k* was multiplied by $-\mu_0/B^2$, which we define as $\chi_\perp$ (see Fig. 3 for details). The dark purple regions highlight where the transverse magnetic susceptibility is largest in field-angle phase space. The values measured below 15 T are set to zero (white) on this plot because dividing by $B^2$ makes the noise diverge at low fields. The metamagnetic phase boundary as determined by the jump in *k* (Fig. 2d) is indicated by teal points outlined in black and terminates at the critical endpoint near $\theta = 59°$ as indicated by the teal star. Note that due to a small misalignment of our crystal, the magnetic field in our measurement traverses a path that is slightly offset from the crystallographic *bc*-plane. Thus we denote the field-angle as $\theta^\star_{bc}$; the angle offsets and the exact path in field-space are discussed further in the SI. The misalignment increases the metamagnetic transition field, while the critical endpoint is lowered into our accessible field range. The line through the metamagnetic transition points is a guide to the eye. The phase boundaries for SC1, SC2, and SC3 are taken from refs. 1 and [28] at 300 mK.

## Magnetotropic measurements

We use the Nanosensors Akiyama A-probe – a commercial silicon microcantilever developed for atomic force microscopy[25,26]. The lever is connected to a quartz tuning fork that electrically drives and detects the resonance frequency of the coupled (fork + lever) oscillators. In DC fields, we use the phase-locked loop (PLL) option of a Zurich Instruments mid-frequency lock-in (MFLI) amplifier to track the resonant frequency and phase of the signal as it evolves with magnetic field and temperature. The experiment operates at the fundamental bending mode of the lever, which ranges from ~35 to 40 kHz. The measured frequency shift is directly proportional to the magnetotropic susceptibility – it is a probe of magnetic anisotropy in the plane of oscillation and field rotation[13].

The magnetotropic measurements were carried out at the National High Magnetic Field Laboratory (NHMFL) in Los Alamos National Laboratory, USA. The tuning fork with the cantilever attached was mounted onto a G10 substrate. The substrate is designed to ensure proper alignment and contact between the fork and the substrate. The G10 substrate is then attached to the stage of the rotator probe. The probe is then inserted into a vacuum-walled stainless steel fridge, and the probe space is pumped down to $10^{-5}$ mbar. The fridge is inserted into a helium-4 cryostat. By introducing a small amount of exchange gas into the sample space, the sample reaches $T = 4$ K (note that atmospheric pressure in Los Alamos is substantially lower than at sea level, and liquid helium is close to 4 K at these altitudes). In order to avoid the magnetic response of the superconductivity itself, a majority of our measurements were performed at 4 K. However, for a few measurements, we pumped on the bath of helium-4 to reach $T = 1.6$ K and measured below $T_c$ (SI Fig. 8).

A frequency scan is performed using a Zurich Instruments MFLI amplifier once the measurement temperature is stable to identify the resonant frequency. We drive the cantilever at its resonance frequency before the field pulse, and then, just before discharging the capacitor bank to the magnet, a trigger signal is sent to stop the cantilever drive, and the cantilever oscillates freely. Throughout the magnetic field pulse, the magnetotropic susceptibility leads to shifts in the oscillation frequency, and the raw data is collected with a digitizer for post-processing. The magnetic field versus time has a full-width at half maximum of ~10 ms. The total duration of the pulse, including its slow decay, is ~100 ms. By the end of the pulse, the amplitude of the resonant frequency, which is also consistently monitored, decays to about 50% of its drive value. With a resonance frequency of roughly 40 kHz, a single lever oscillation cycle occurs in 25 $\mu$s. Several oscillations are analyzed over small time windows of 100-250 $\mu$s to find the resonance frequency, and the windows are stepped by 20 $\mu$s to generate each data point. The zero-field frequency is always remeasured between measurements taken at new field angles (i.e., after rotating the sample).

## Data availability

The datasets generated during the current study are available at the ISTA Research Explorer repository (https://doi.org/10.15479/AT-ISTA-21174).

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

## Acknowledgements

We appreciate technical support from Salvatore Bagiante, Evgeniia Volobueva, Lubuna Shafeek, Ali Bangura, and Zoltán Köllö, and scientific discussions with Daniel Agterberg, Johnpierre Paglione, Qimiao Si, Josephine Yu and Yue Yu. V.Z., A.N., M.N., and K.A.M. acknowledge funding received from the European Research Council (ERC) under the European Union's Horizon 2020 research and innovation programme (TROPIC-101078696). V.Z., A.N., M.N., and K.A.M. thank the ISTA Nano-fabrication Facility for technical support. B.J.R. acknowledges funding from the Office of Basic Energy Sciences of the United States Department of Energy under award number DE-SC0020143 for data analysis and writing. The National High Magnetic Field Laboratory is supported by the National Science Foundation through NSF/DMR-2128556*, the State of Florida, and the U.S. Department of Energy. A.S. acknowledges support from the DOE/BES "Science of 100 T" grant. A.S. thanks Downtown Subscription in Santa Fe, NM, for their patience in hosting him. Sample preparation and characterization were supported by the NSF through DMR-2105191.

## Author contributions

B.J.R. and K.A.M. conceived of the experiment; S.K.L., C.E.F., and N.P.B. prepared and characterized the samples; V.Z., A.N., M.N., and A.S. performed the experiments and analyzed the data. V.Z., B.J.R., and K.A.M. wrote the manuscript with input from all co-authors.

## Competing interests

The authors declare no competing interests.
