## [Transparent Peer Review file · Nature Communications]

Giant transverse magnetic fluctuations at the edge of re-entrant superconductivity in UTe_2

Corresponding Author: Professor Kimberly Modic

Version 0:

Reviewer comments:

Reviewer #1

(Remarks to the Author)

The manuscript by Zambra et al. reports evidence for transverse magnetic fluctuations near the edge of the field-induced reentrant superconductivity in UTe_2 by using a recently developed new technique, namely the magnetotropic susceptibility measurements. The work provides important insights into understanding the mechanism of this exotic reentrant superconducting state and therefore deserves publication. Before I recommend it for publication in Nature Communications, I would like to ask the authors to address/clarify the following issues:

1. In the abstract, the authors pointed out that "...magnetization measurements show no sign of strong fluctuations." However, evidence for clear antiferromagnetic (AF) spin fluctuations was previously detected by neutron scattering, see Chunruo Duan et al., Nature 600, 636–640 (2021). The authors should describe it more precisely. How do the AF fluctuations evolve under a magnetic field? Which type of spin fluctuations do the observed giant transverse magnetic fluctuations belong to? Can the authors provide information or a discussion on it?
2. Typically, superconductivity is favored by strong spin fluctuations if it is driven by magnetic interactions. However, according to Fig. 4, it seems that superconductivity and magnetic fluctuations are repelling each other, superconductivity being far away from B^* . In order to reveal the possible relationship between B^* and B^{max} , and therefore to elucidate the link between superconductivity and the transverse magnetic fluctuations, measurements along other field- and sample-orientations are needed.
3. Sample #1 (main text) and sample #3 (SI) show rather different behavior near B/c . What is origin of the jump near 20T? The authors attribute it to a sample misalignment, but this is just a speculation and more detailed measurements are needed to elucidate this problem.

One minor point: According to the definition of $\chi_{\{ij\}}$ in Eq. 1, j represents the external field orientation. Therefore, the label of $\chi_{\{aa\}}(B||c)$ or $\chi_{\{bb\}}(B||c)$ is probably misleading. Is there any better expression?

Reviewer #2

(Remarks to the Author)

Zambra et al present a high magnetic field study of the magnetotropic susceptibility of the unconventional superconductor UTe_2 . This measurement technique is sensitive to transverse spin fluctuation modes, which are difficult to probe by other techniques. Performing measurements sweeping the magnetic field orientation through two orthogonal rotation planes, a pronounced 'softening' of the magnetotropic susceptibility is reported close to the crystallographic c -axis, which is the hard magnetic axis at high fields. The authors attribute this observation to pronounced transverse spin fluctuations, which may have important consequences for the manifestation of exotic magnetic field-induced superconductivity in this material.

This is an interesting study that provides an important new perspective on the anomalous high magnetic field superconducting phases of UTe_2 . Overall, the data appear reasonably solid, and the conclusions – albeit limited in scope – are well justified. However, I have some concerns with the manuscript that should be addressed before publication:

1. My major concern regards the knowledge of sample alignment. In Fig. 2d the authors report that a 3 degree rotation abruptly marks the end of the field polarized state. In Fig4 this observation is used to justify the identification of a point B^* at around 50 T, that the authors claim is a critical end point, and a discussion with the results of ref. 13 is given. However, in ref. 13 it appears for the bc rotation plane the CEP lies closer to 75 T (fig. 20). Taking the bc metamagnetic transition points reported in doi.org/10.1073/pnas.2403067121 and comparing with the present work, I roughly get the attached file. Clearly there is a large disagreement here. However, the metamagnetism of UTe₂ has been widely studied by many groups, and should be $\sim 35T$ for B/b , whereas in the present work it is clearly higher, and remains systematically higher for inclined rotation angles. The most plausible cause is that there is some a-axis misalignment present here, which pushes H_m higher at '0 degrees'. As reported in doi.org/10.1126/science.adn7673 H_m rises quicker out of the bc plane (i.e., when some nonzero a-axis field component is present). The authors give some discussion of sample misalignment in the supplement, but claim that the data in the main figures are from a well aligned sample. This is clearly not the case. The authors should do a much more thorough job of identifying the actual rotation plane measured here. At present, it is hard to really trust that 'B_max' is really where they claim it is, as the knowledge of angles is clearly not very good. Actually, it seems they are off somewhere in H_{abc} space, which appears to match the conclusions of ref. 13. It should be possible to work out the true rotation using the $\sin^2 \sin^4$ eqn of ref 7. I think in Fig2 what is actually happening is that H_m has just gone above the accessible field range of this experiment, and not actually got close to the CEP. It is also perplexing that Fig. S4 shows H_m suddenly disappear, after a reported mere 1 degree of rotation from 78 to 77deg. Have these angles been checked with the pickup coils of the rotation probe?! Sadly, there seems to be some major mistake(s) here, which really detract from the credibility of the paper.

2. In line 25 of the abstract, and on line 42 of page 2, it is asserted that magnetic fluctuations have not previously been observed at high fields in UTe₂. However, this is not true: these were reported in doi.org/10.1103/PhysRevB.109.L140502 and doi.org/10.1103/PhysRevLett.131.226503. Furthermore, a model for the pairing of SC2 based on metamagnetic fluctuations was proposed in doi.org/10.1073/pnas.2403067121. These prior results should be discussed, and the wording amended given that the present work is not the first study to show evidence for strong magnetic fluctuations in UTe₂.

3. This technique for studying magnetotropic susceptibility will likely not be particularly well known or understood by many readers, such as myself. It is therefore hard to really understand or appreciate what this 'softening' really refers to. The form of eqn 1 looks similar to magnetic torque, where signal sizes can be misleading due to a dependence on $\sin(\theta)$. For example, in fig 1 of ref4 the size of the transition in UTe₂ looks as though it's growing upon rotating further away from the b-axis. But in reality, it is shrinking in magnitude, but it's the raw data plotted rather than dividing by $\sin(\theta)$. I wonder about the present manuscript fig2d – is it really 'softening' or does this need dividing by $\sin(\theta)$? If not, why not? The jump at H_m indeed seems to grow like the torque in ref4, which I presume is just a trivial result of a cross product term. Some discussion of this is warranted.

4. In Fig. 1c there is a change of sign of k_a as a function of B. Does this physically correspond to something interesting? This is not expected in the calculation – is this also perhaps a feature of sample misalignment?

5. In Fig1 refs 4 & 11 are used to construct the phase diagram, whereas in Fig4 refs 7 & 5 are used. This should be made consistent. In fact, to my knowledge the appropriate, most up to date references to use for this would be doi.org/10.7566/JPSJ.93.123702 doi.org/10.1073/pnas.2403067121 and doi.org/10.1073/pnas.2422156122

6. Why is a c-axis curve plotted in fig2b but not 2d? This looks odd

7. Refs 13 and 14 appear to be the same paper, which now has doi: doi.org/10.1103/PhysRevX.15.021019

In summary, I think this is an interesting result that provides important new information about the unusual high field physics of UTe₂. If the authors can provide a more thoughtful assessment of the true sample orientation (or at least some reasonable range of uncertainty), I would support its publication in Nat Comm.

Reviewer #3

(Remarks to the Author)

The manuscript presents an experimental study of the exotic superconductor UTe₂, using state-of-the-art magnetotropic susceptibility measurements in high fields. The results reveal strong (ferromagnetic) spin fluctuations, particularly near the quantum critical endpoint (QCEP) of a first-order phase transition driven by external magnetic fields, which is conjectured to be the "pairing glue" for such high-field, re-entrant superconductive phase. The findings are of significant interest, and the technical methodology is sound, aside from some minor typos (noted below). I am therefore inclined to recommend publication of this interesting work in Nature Communications, given the following comments/questions are well addressed in their revised manuscript.

1) I understand Eq. (1) is an exact expression. Therefore, if one has access to both the longitudinal magnetization (the induced magnetization along the field direction) and the differential transverse susceptibility (perpendicular to the axis n and the magnetic field B), then it should be possible to exactly reconstruct the magnetotropic susceptibility. Is this understanding correct?

If so, why do the purple dashed lines in Fig. 2(a, b) deviate from the k_n curves? Could it be that they were estimated using not the correct transverse susceptibility but only the longitudinal magnetization? I am somewhat confused by this

discrepancy.

2) In Fig. 2d of the main text (as well as in Supplementary Fig. 4b), a jump is observed in the magnetotropic susceptibility χ_a , for large angles ($\theta > 59^\circ$) and at approximately 40–50 T. This jump is used to identify the metamagnetic transition. However, it is not entirely clear to me whether this jump originates from the first or the second term in Eq. (1).

The case at $\theta = 59^\circ$ is near the QCEP, where the jump/dip is most pronounced. Since the longitudinal component of magnetization is not expected to exhibit a jump at the QCEP — given that it constitutes a continuous quantum phase transition — I suspect that the jump in χ_a arises primarily from transverse fluctuations. Is this interpretation correct? In any case, additional discussion on this point seems warranted.

Furthermore, could authors clarify how the transverse susceptibility χ_\perp presented in the contour plot Fig. 4 was obtained?

3) A particularly intriguing finding is the QCEP and the strong FM fluctuations detected via χ_n in its vicinity. This discovery establishes a quantum analogue of the critical endpoint (CEP) and supercritical states in a classical liquid-gas phase diagram. Relatedly, in the ground-state phase diagram of Ising model under longitudinal and transverse fields, a first-order transition line terminates at a QCEP, giving way to a quantum supercritical regime with strong FM fluctuations.

Furthermore, the measured deviation of B_{\max} from B^* may also arise from finite-temperature effects, as the experiments were conducted at 4 K (well above the SC transition). The influence of a QCEP at finite temperature has also been explored within the quantum Ising model, characterized by large susceptibility (corresponding to the transverse susceptibility here) amongst other thermodynamics (see arXiv:2410.22236).

minor points:

i) Caption of Fig. 1, “ $\chi_b(B||b) = B (M_b - \dots)$ ” seems a typo, should it be “ $\chi_a(B||b)$ ”?

ii) In Figure 1a, the metamagnetic transition forms a continuous boundary. This appears to contrast with Figure 4, where a QCEP terminates the first-order transition line.

iii) References [7] and [14] (currently cited as arXiv preprints) have since been published. The published versions can be cited instead.

Version 1:

Reviewer comments:

Reviewer #1

(Remarks to the Author)

In my previous report, I requested further measurements to clarify some problems (similar concerns were also raised by other referees). The authors failed to perform these measurements and instead provided analyses of the sample misalignment based on the comparison with some previously published results. I can understand that such measurements are time consuming. On the other hand, their arguments seem reasonable. Therefore, I'm fine if the editor is going to publish this manuscript in NC.

BTW, the sequence of the references should be sorted out.

Reviewer #2

(Remarks to the Author)

The authors have performed an admirable job in patiently responding to each of the referees' questions. The efforts to quantify and fairly present the angular misalignments are very robust. I have a few small suggestions listed below, which could help further strengthen the paper's impact:

Firstly, I spotted a small typo in the supplement on page 3: “according the angle” should be “according to the angle”.

The field of UTe₂ still seems to be moving quickly, and there are a number of recent results that the (still rather short) discussion section might consider expanding on. Notably, in Lewin et al PRB 110, 184520 (2024) and Wu et al arXiv:2503.11362 anomalous features in the UTe₂ high field phase landscape have been observed in conductivity measurements. These could possibly be related to the present work(?)

The introduction briefly mentions the disorder sensitivity (or lack thereof) in the Frank et al ‘orphan’ paper. However, a fascinating recent study looking at thorium-substitution (Moir et al, PNAS e2521261122 2025) has shown that SC1 can be preserved while SC3 is suppressed! This points to a leading role for dimer?

Also, clear signatures of non-Fermi liquid behavior have been resolved in arXiv:2505.12131 and arXiv.2505.09351. These seem to correspond well to where the B^* star is located in Fig 4 of the present study. This seems like very nice corroboration between 3 independent groups (including this work).

In summary, I find the additions to the supplement to strongly enhance the credibility of the paper, which provides a very interesting new perspective on the high magnetic field properties of the enigmatic superconductor UTe₂. I list the above

suggestions merely for the authors' consideration, and strongly support the paper's publication in Nat Comm.

Reviewer #3

(Remarks to the Author)

After reading the revised manuscript and the authors' detailed response to the reviewers' comments, I find that the authors have addressed the points carefully, and the overall quality of the work has been improved. While it is still not entirely clear to me (i) the origin of the jump in Fig. 2d, and (ii) why the superconductivity region appears to repel the strong transverse ferromagnetic fluctuations, I believe the experimental results are substantial and deserve publication in their current form.

I have only one minor point for the authors' consideration. Although the description of B_{\max} has been removed, the remaining discrepancy might still originate from quantum fluctuation effects. The phase boundary in Fig. 4 is determined at a finite temperature. If one imagines lowering the temperature further, this boundary should still take the form of a first-order transition line that terminates at a critical endpoint (CEP). The zero-temperature limit of this CEP would constitute a quantum critical endpoint (QCEP). This QCEP is expected to be located at a field distinct from B^* and could strongly influence low-temperature magnetothermodynamics — including the magnetotropic response measured here.

REVIEWER COMMENTS

Reviewer #1 (Remarks to the Author):

The manuscript by Zambra et al. reports evidence for transverse magnetic fluctuations near the edge of the field-induced reentrant superconductivity in UTe₂ by using a recently developed new technique, namely the magnetotropic susceptibility measurements. The work provides important insights into understanding the mechanism of this exotic reentrant superconducting state and therefore deserves publication. Before I recommend it for publication in Nature Communications, I would like to ask the authors to address/clarify the following issues:

1. In the abstract, the authors pointed out that "...magnetization measurements show no sign of strong fluctuations." However, evidence for clear antiferromagnetic (AF) spin fluctuations was previously detected by neutron scattering, see Chunruo Duan et al., Nature 600, 636–640 (2021). The authors should describe it more precisely. How do the AF fluctuations evolve under a magnetic field? Which type of spin fluctuations do the observed giant transverse magnetic fluctuations belong to? Can the authors provide information or a discussion on it?

Thank you for pointing out that the abstract was not accurate. We have modified it to include reference to the fact that, at least at fields below 11 tesla, neutrons observe that the spin susceptibility peaks at an AFM wavevector.

Our measurements probe $q=0$ and therefore are most sensitive to ferromagnetic fluctuations. While our data suggest evidence for strong ferromagnetic fluctuations, we cannot rule out fluctuations from another q -vector that are broad in momentum. Our report of large ferromagnetic fluctuations at high fields is consistent with a recent NMR study [PRL 131, 226503 (2023)], which finds an enhancement of both $1/T_1$ and $1/T_2$ with increasing magnetic field with no sign of an anomaly at a select wavevector. They also observe a uniform Knight shift with increasing field and a very large ratio of T_1/T_2 , both of which suggest a uniform (ferromagnetic) fluctuating spin component.

More details about the type of fluctuations observed are now included in the abstract, introduction and discussion. We also added citations to the relevant works that we missed.

2. Typically, superconductivity is favored by strong spin fluctuations if it is driven by magnetic interactions. However, according to Fig. 4, it seems that superconductivity and magnetic fluctuations are repelling each other, superconductivity being far away from $B^{\{max\}}$. In order to reveal the possible relationship between B^* and $B^{\{max\}}$, and therefore to elucidate the link between superconductivity and the transverse magnetic fluctuations, measurements along other field- and sample- orientations are needed.

We agree that further measurements will be important to fully understand the link between the observed spin fluctuations and superconductivity. Our focus in this work is on the main experimental finding – the large transverse fluctuations that we observed for a broad range of angles in UTe₂. We confirmed this observation by doing two separate high-magnetic field experiments where we measured in two different crystal planes on three different samples, one which cut using the focused ion beam and two coming from a bulk crystal.

It was also surprising to us that the region of large transverse fluctuations appears to repel the superconducting phases. One possible explanation is that, while transverse fluctuations provide a possible superconducting "glue", other ingredients are also necessary for superconductivity, such as a large density of states and magnetic fluctuations that are "matched" to the Fermi surface geometry. Perhaps some or all of these factors are not supportive of SC even in the observed region of the large fluctuations. See the work by J Yu, et. al. [[arXiv:2504.07088](https://arxiv.org/abs/2504.07088)] where a Zeeman field pushes the edge of a band past the chemical potential, resulting in a large density of states. This large density of states then helps the pairing mechanism, whatever it is, to produce a high-field superconducting state. We have added this to the discussion.

It is important to remember that while the orientation of the crystal on the lever selects out just one component of the magnetic susceptibility tensor that is transverse to the applied magnetic field, there is an entire plane that exists for each field direction where the magnetotropic susceptibility could be measured transversely. It's possible that any other transverse component of the magnetic susceptibility tensor could be largest when magnetic field is applied near the critical endpoint (e.g. the maximum transverse susceptibility may be closer to B^* in some *other* transverse direction. While we would like to explore more field orientations in various crystal planes, there is a very large phase space to be explored, not only due to the complexity of the field-angle phase diagram in UTe₂, but also due to the complexity of the measurement. We tried to clarify this point in the discussion.

3. Sample #1 (main text) and sample #3 (SI) show rather different behavior near B/c . What is origin of the jump near 20T? The authors attribute it to a sample misalignment, but this is just a speculation and more detailed measurements are needed to elucidate this problem.

The main difference between all three samples is a slightly different plane of rotation for the magnetic field. This produces two key differences in the data sets: the first is that the data in samples #2 (ac-plane) and #3 (bc-plane) in the SI both show a softening that is smaller in magnitude across the full field range compared to sample #1 in the main text; the second is that the positions of the features as a function of magnetic field are at slightly different fields and angles. For example, as pointed out by the reviewer, the data in the SI shows a jump near 20-25 T (largest for field along the c-axis) that is dependent upon field angle, and which looks like a second-order phase transition.

To illustrate what a jump in the magnetotropic susceptibility means, let's first discuss the second-order phase transition (critical endpoint) near the metamagnetic transition. At higher fields, the first-order metamagnetic transition is expected to be bounded by a line of second-order critical endpoints [Wu *et. al.*, PRX 15, 021019 (2025)], and we cross this line of critical endpoints twice in our experiments – both in the SI and in the main text for the bc-plane data, at ~ 78 and ~ 59 degrees, respectively. The critical endpoints were previously identified by a reduction (to zero) in the jump of the magnetization at the metamagnetic transition. Our measurements are different in that they probe a second-derivative of the free energy (whereas magnetization is a first derivative), and are therefore expected to show discontinuous behavior (i.e. a jump) across *second-order* phase transitions (as well as, in general, jumps at first order transitions if the susceptibility changes on either side of the transition). The discontinuity observed in our measurement at the first-order metamagnetic transition grows in amplitude before abruptly terminating near the same field-angle where the critical endpoint is suggested by Wu *et. al.* This is consistent with the transition becoming less first-order-like and more second-order-like approaching the endpoint.

Now, connecting this back to the referee's question of why we see a jump near 20 T in the SI data: this appears to be either a second order or weakly first order phase transition. As shown by Wu *et al.*, the position in field of the critical endpoint of the metamagnetic transition depends strongly on the field orientation. Thus, it is possible that we are observing the critical end point of the metamagnetic transition (keep in mind that the misalignment is larger in the SI sample, and we are not measuring in a direction probed by Wu *et al.*). Another possibility is that there is a *distinct* second order phase transition in the phase diagram of UTe₂, which would not be unusual given the complex nature of this material. In order to say anything definitive about this apparent phase transition, we will need future studies that measure in many more field orientation planes. However, because it bears no weight on the conclusions drawn based on the softening of the magnetotropic coefficient, we chose not to discuss this feature in the main text. We would like to revisit the phase boundaries and these signatures in a future study.

In the revised version, we discuss the sample misalignment in detail in Section II of the SI, which we hope will provide the necessary clarity for astute readers, while keeping the key findings as simple as possible. **We also incorporated the suggested changes to the phase diagram in Figure 4** (related to this point and point #5 made by reviewer 2 below), which now more accurately reflects the field directions that were measured.

To summarize how we have addressed sample misalignments in the SI: the higher metamagnetic transitions observed in our measurements as compared to those published previously suggest a non-zero component of magnetic field along the a-axis in our bc-plane measurements. We use our measurements of the positions of the metamagnetic transition in field-angle space, along with the comprehensive mapping of the critical fields determined by Lewin *et. al.* [Science 389;6759 pp 512-515 (2025)], to determine the angle offsets. We find that a rotation of the sample around its c-axis by ~ 8 degrees, followed by a tilt of the sample by ~ 4 degrees, maps our metamagnetic transitions for the data presented in Figure 2d of the main text to those of Lewin *et al.* This small misalignment is difficult to avoid because we work with very small samples (hundreds of nanograms - more details are given in the response to Referee #2). The angle offsets explain both why we observe slightly higher values of the metamagnetic transition, as well as why the critical endpoint is within our field range.

We have also worked out the misalignments for the bc-plane measurements for the data shown in the SI and also included this information in Section II of the revised SI. Qualitatively, we feel that the agreement between the two data sets, especially given that the data is taken on different samples and with different misalignments, is quite remarkable; the main finding of the experiment is the large negative response of the magnetotropic susceptibility for a broad range of field angles that is truncated in the bc-plane by the metamagnetic transition.

One minor point: According to the definition of $\chi_{\{ij}}$ in Eq. 1, j represents the external field orientation. Therefore, the label of $\chi_{\{aa\}}(B||c)$ or $\chi_{\{bb\}}(B||c)$ is probably misleading. Is there any better expression?

We think that the label is accurate, but that it was not explained clearly in the text. By oscillating the cantilever, the orientation of the sample in the magnetic field also oscillates. This oscillates the component of the magnetic field perpendicular to the applied field, probing the susceptibility in this direction. Thus the ∂B_j appearing in the definition of χ after equation 1 is the *oscillating* field component, not the large applied field (which is the vector B). **Around line 75 in the revised text, we have now added text to clarify this point.**

Thank you for your time and thoughtful reading of our manuscript.

Reviewer #2 (Remarks to the Author):

Zambra et al present a high magnetic field study of the magnetotropic susceptibility of the unconventional superconductor UTe₂. This measurement technique is sensitive to transverse spin fluctuation modes, which are difficult to probe by other techniques. Performing measurements sweeping the magnetic field orientation through two orthogonal rotation planes, a pronounced 'softening' of the magnetotropic susceptibility is reported close to the crystallographic c-axis, which is the hard magnetic axis at high fields. The authors attribute this observation to pronounced transverse spin fluctuations, which may have important consequences for the manifestation of exotic magnetic field-induced superconductivity in this material.

This is an interesting study that provides an important new perspective on the anomalous high magnetic field superconducting phases of UTe₂. Overall, the data appear reasonably solid, and the conclusions – albeit limited in scope – are well justified. However, I have some concerns with the manuscript that should be addressed before publication:

1. My major concern regards the knowledge of sample alignment. In Fig. 2d the authors report that a 3 degree rotation abruptly marks the end of the field polarized state. In Fig4 this observation is used to justify the identification of a point B^* at around 50 T, that the authors claim is a critical end point, and a discussion with the results of ref. 13 is given. However, in ref. 13 it appears for the bc rotation plane the CEP lies closer to 75 T (fig. 20). Taking the bc metamagnetic transition points reported in doi.org/10.1073/pnas.2403067121 and comparing with the present work, I roughly get the attached file. Clearly there is a large disagreement here. However, the metamagnetism of UTe₂ has been widely studied by many groups, and should be ~ 35 T for $B//b$, whereas in the present work it is clearly higher, and remains systematically higher for inclined rotation angles. The most plausible cause is that there is some a-axis misalignment present here, which pushes H_m higher at '0 degrees'.

This is correct – the most natural explanation for our higher H_m that increases more sharply with field angle is an a-axis misalignment in our measurements. **According to suggestions from all reviewers, we have included a more thorough analysis of the sample misalignments in Section II of the SI** (more details given below). Prior to the experiments, we placed much emphasis on aligning crystals to the best of our ability, but their small sizes (of order 10's of microns per side) make it difficult to use the usual scattering techniques to detect the crystal axes after mounting the samples. Furthermore, in pulsed field measurements, even a small initial misalignment can be amplified due to the stacking of several substrates that are needed for rotation studies.

As reported in doi.org/10.1126/science.adn7673 H_m rises quicker out of the bc plane (i.e., when some nonzero a-axis field component is present). The authors give some discussion of sample misalignment in the supplement, but claim that the data in the main figures are from a well aligned sample. This is clearly not the case. The authors should do a much more thorough job of identifying the actual rotation plane measured here. At present, it is hard to really trust that ' B_{max} ' is really where they claim it is, as the knowledge of angles is clearly not very good. Actually, it seems they are off somewhere in H_{abc} space, which appears to match the conclusions of ref. 13. It should be possible to work out the true rotation using the $\sin^2 \sin^4$ eqn of ref 7.

To summarize the analysis done in Section II of the SI for your convenience here, we found that sample #1 (shown in the main text figures) was rotated around the c-axis by ~ 8 degrees, and then tilted by ~ 4 degrees. We now explain how we reconstruct our experimental rotation plane by fitting our experimental H_m to that from reference 7. In SI Figures 4 and 5, we show our H_m data alongside the expected H_m data based on reference 7 for the "corrected angles" that we measure in our experiments—the agreement is good, indicating that there is reproducibility in the shape of the phase boundary once the rotation angles are corrected .

The offset angles we extract from this analysis are in broad agreement with what was previously written in the SI, which read "Alignment of our samples, which are only 100's of nanograms, is a technical challenge. As all features in UTe₂ are highly-anisotropic, one expects that slight misalignments may lead to variation in the positions of features observed with angle. Assuming 10 degree offsets, our data are consistent with that of Lewin et al. [35]." **Now, we have incorporated the full analysis and more discussion about misalignments for both experiments; sample #1 in the main text, and samples #2 and #3 in the SI. We have also clarified in**

the phase diagram of Figure 4 that our measurements were not done precisely in the bc-plane, with the exact rotation plane shown in the SI. We hope this helps to avoid future confusion.

In Figure 4, we decided to remove the label B_{\max} from the phase diagram. B_{\max} was meant to signal the region where the observed transverse fluctuations were maximum, but we think that it incorrectly suggested a critical point, which is particularly confusing in the context of UTe₂. **We also adjusted the color scale in Figure 4** because we believe that it falsely over-emphasized the small region of the phase diagram where B_{\max} was previously labelled. The main experimental take-away should be that the magnetotropic softening occurs across most of the bc-plane (and some of the ac-plane) at high fields, excluding the field-polarized phase. Now, the new color scheme in Figure 4 highlights this experimental fact better; the cyan/blue regions correspond to the typical size of the longitudinal susceptibility in the linear regime, while the purple region shows the region of large transverse susceptibility, with deep purple representing where it is maximum. We believe that this more accurately emphasizes the main finding.

I think in Fig2 what is actually happening is that H_m has just gone above the accessible field range of this experiment, and not actually got close to the CEP. It is also perplexing that Fig. S4 shows H_m suddenly disappear, after a reported mere 1 degree of rotation from 78 to 77deg. Have these angles been checked with the pickup coils of the rotation probe?! Sadly, there seems to be some major mistake(s) here, which really detract from the credibility of the paper.

We do not believe that H_m has moved out of the accessible field range in Figure 2d. Extrapolating the position of H_m versus θ suggests that, if there was no critical endpoint, then the MM transition at $\theta = 56$ degrees should be approximately 48 tesla. This is well within our accessible field range, and yet the transition is absent at 56 degrees. This same behavior was also observed in sample #3 in the SI.

This disappearance of the MM transition is expected based on the mapping of the critical endpoint of the MM transition based on Wu, *et. al.* [Fig 1 c in reference 13]. There, the sample is mounted with the magnetic field 17 degrees from b towards c, and then the field is rotated toward a. They show that the jump in the MM transition disappears upon further rotating towards the a-axis by 19 degrees. This is expected at the critical endpoint of a second order phase transition—first derivatives of the free energy, like magnetization, go to zero at the critical end point. In contrast, the magnetotropic coefficient is a second derivative of the free energy and therefore we expect the jump to increase approaching the critical endpoint. This is exactly what we observe. The fact that H_m disappears abruptly is consistent with a line of first-order metamagnetic transitions in field-angle space abruptly terminating at a second-order critical endpoint. In our measurements in Figure 2d, we have a smaller tilt towards the a-axis compared to Wu, *et. al.*, which actually reduces the critical endpoint such that we access it within our available field range. **We now explain how our measurements and magnetization are consistent with the expected behavior for first- and second-order phase transitions in the results section.**

Our angles are initially determined by measuring the rotation stage angle using a pick-up coil. We then correct for misalignment between the rotation stage and the sample by examining the low field angle dependence, which behaves as $\cos^2\theta$ in magnetotropic susceptibility measurements.

We also would like to note that the data in Fig 20 of reference 13 has been scaled according to the size of a jump at the metamagnetic transition for one angle. Measurements in both Figure 1 and Figure 20 of reference 13 are taken on different samples at different magnet labs, each with their own sources of small samples misalignments, and then they are rescaled to match one another. Any small misalignment could contribute significantly to error in this rescaling. In the revised version, we tried to be more transparent about the degrees of misalignment encountered in our measurements, while not tying ourselves too closely to misalignments possibly introduced by the measurements of others. With that being said, we believe that the agreement across measurements with Wu *et. al.* [PRX 15, 021019 (2025)], Lewin *et. al.* [Science 389;6759 pp 512-515 (2025)], and others are quite good.

To summarize, we have now calibrated the sample misalignment as discussed in response to the previous point. Our phase diagram, including the location of the critical endpoints, is in good agreement with Wu *et. al.*, PRX 15, 021019 (2025).

2. In line 25 of the abstract, and on line 42 of page 2, it is asserted that magnetic fluctuations have not previously been observed at high fields in UTe₂. However, this is not true: these were reported in doi.org/10.1103/PhysRevB.109.L140502 and doi.org/10.1103/PhysRevLett.131.226503. Furthermore, a model for the pairing of SC2 based on metamagnetic fluctuations was proposed in doi.org/10.1073/pnas.2403067121. These prior results should be discussed, and the wording amended given that the present work is not the first study to show evidence for strong magnetic fluctuations in UTe₂.

Thank you for pointing this out. **We have amended the wording, and we hope that all of the relevant references to previous reports of fluctuations are now included in the abstract and introduction.** The additional insight provided by our measurements is that the high-field behavior is dominated by ferromagnetic, transverse fluctuations.

3. This technique for studying magnetotropic susceptibility will likely not be particularly well known or understood by many readers, such as myself. It is therefore hard to really understand or appreciate what this ‘softening’ really refers to. The form of eqn 1 looks similar to magnetic torque, where signal sizes can be misleading due to a dependence on $\sin(\theta)$. For example, in fig1 of ref4 the size of the transition in UTe₂ looks as though it’s growing upon rotating further away from the b-axis. But in reality, it is shrinking in magnitude, but it’s the raw data plotted rather than dividing by $\sin(\theta)$. I wonder about the present manuscript fig2d – is it really ‘softening’ or does this need dividing by $\sin(\theta)$? If not, why not? The jump at H_m indeed seems to grow like the torque in ref4, which I presume is just a trivial result of a cross product term. Some discussion of this is warranted.

Thank you for pointing out potential confusion for others. We referred to the negative response of the magnetotropic susceptibility at high fields as “softening”, mainly because this behavior is observed as a negative shift in the resonant frequency of the cantilever. It becomes softer because the lever prefers to move in the direction transverse to the applied field—a sign that the magnetic field is applied along a magnetically-hard axis.

It is correct that magnetic torque scales as $\sin(2\theta)$ in the linear response regime (i.e. when the magnetization is linearly proportional to the applied field). In general (beyond linear response), torque will contain higher harmonic contributions (e.g. $\sin(4\theta)$, $\sin(6\theta)$, etc). In the case of magnetotropic, in the linear response regime, the scaling factor is instead $\cos(2\theta)$. Again, beyond linear response (which is highly relevant for UTe₂ in high fields), there are contributions from $\cos(4\theta)$, etc.

The distinction in symmetry between torque—odd in angle around high symmetry directions—and magnetotropic—even in angle around high symmetry directions—is important because it implies that the maximum magnetotropic response occurs along the principal magnetic axes, rather than in between them as is the case with torque. In Figure 2 of our paper, at low magnetic fields (<10 T), our data evolves with angle as $\cos(2\theta)$ in the ac-plane. The $\cos(2\theta)$ response can be divided out to calibrate the magnitude of the magnetic anisotropy in the linear regime. However, in the bc-plane, the b- and c-components of the magnetic susceptibility are almost the same (i.e. the curves are basically flat in Figure 2d below ~10 T), and so in this case the linear response is almost absent—almost the entire response is beyond linear order. Thus it is not particularly helpful to divide by $\cos(2\theta)$ in this case.

Specifically to the referees point—because magnetotropic evolves as $\cos(2\theta)$, and not $\sin(2\theta)$, the growth of the jump at H_m rotating away from the b axis is actually *diminished* by these angular factors. **We added more discussion related to how the jump at the metamagnetic transition evolves in the magnetotropic susceptibility upon approaching the critical endpoint.**

4. In Fig. 1c there is a change of sign of k_a as a function of B. Does this physically correspond to something interesting? This is not expected in the calculation – is this also perhaps a feature of sample misalignment?

To be clear, both the calculated and measured k_a in 1c are negative until the metamagnetic transition. If the referee is referring to the fact that both curves become positive at the metamagnetic transition: this is because the magnetic susceptibility components for fields along both b and c cross at the metamagnetic transition, as shown in Figure 1b.

However, the referee may instead be referring to the fact that the measured curve has a larger response and crosses the calculated curve near 20 tesla. Indeed, some part of this may be due to sample misalignment. However, it is worth noting that, in the calculated curve, only longitudinal fluctuations are accounted for (i.e. the calculated curve uses only the longitudinal magnetization curves and their derivatives from Figure 1b). Thus the departure of the measured curve from the calculated curve may indicate the importance of transverse susceptibility even at fields less than 20 tesla.

5. In Fig1 refs 4 & 11 are used to construct the phase diagram, whereas in Fig4 refs 7 & 5 are used. This should be made consistent. In fact, to my knowledge the appropriate, most up to date references to use for this would be doi.org/10.7566/JPSJ.93.123702 doi.org/10.1073/pnas.2403067121 and doi.org/10.1073/pnas.2422156122

For consistency, we have now constructed the phase diagrams in Figures 1 and 4, both based on Lewin *et al.* [Science 389:6759 pp 512-515 (2025)] and Ran *et al.* [Nat. Phys.15:125-1254, (2019)]. The main difference between the two phase diagrams now is that the angles in Figure 4 have been corrected (based on the critical

fields measured by Lewin *et. al.* and Ran *et. al.*) to accurately reflect the planes that were measured in our experiment. Specifically, **we relabelled the θ -axis in Figure 4 and adjusted the figure caption to reflect that the measurement plane does not lie perfectly within the crystallographic bc-plane.**

The references mentioned above provide phase diagrams for “ultra-clean” samples with a higher T_c than those used in our experiments. With these higher T_c s come higher critical fields for SC1 and SC2. Thus, we compared our results to Lewin *et. al.* because T_c 's of their samples are similar to ours. Despite these differences, it is worth pointing out that the critical fields of both the re-entrant SC phase and metamagnetism appear to be much less sensitive to T_c (and thus variations in disorder).

6. Why is a c-axis curve plotted in fig2b but not 2d? This looks odd

Thanks for pointing this out. The data in Figure 2b is the same (only divided by field) as that shown in Figure 2d (i.e. they are both measured with magnetic field 6 degrees away from the c-axis). **We have now corrected the label in Figure 2b to indicate that the field is 6 degrees away from c and not exactly aligned along c.**

7. Refs 13 and 14 appear to be the same paper, which now has doi: doi.org/10.1103/PhysRevX.15.021019

Thanks. This has been corrected.

In summary, I think this is an interesting result that provides important new information about the unusual high field physics of UTe₂. If the authors can provide a more thoughtful assessment of the true sample orientation (or at least some reasonable range of uncertainty), I would support its publication in Nat Comm.

Thank you for carefully considering our manuscript.

Reviewer #3 (Remarks to the Author):

The manuscript presents an experimental study of the exotic superconductor UTe₂, using state-of-the-art magnetotropic susceptibility measurements in high fields. The results reveal strong (ferromagnetic) spin fluctuations, particularly near the quantum critical endpoint (QCEP) of a first-order phase transition driven by external magnetic fields, which is conjectured to be the “pairing glue” for such high-field, re-entrant superconductive phase. The findings are of significant interest, and the technical methodology is sound, aside from some minor typos (noted below). I am therefore inclined to recommend publication of this interesting work in Nature Communications, given the following comments/questions are well addressed in their revised manuscript.

1) I understand Eq. (1) is an exact expression. Therefore, if one has access to both the longitudinal magnetization (the induced magnetization along the field direction) and the differential transverse susceptibility (perpendicular to the axis n and the magnetic field B), then it should be possible to exactly reconstruct the magnetotropic susceptibility. Is this understanding correct?

Yes, this is indeed correct.

If so, why do the purple dashed lines in Fig. 2(a, b) deviate from the k_n curves? Could it be that they were estimated using not the correct transverse susceptibility but only the longitudinal magnetization? I am somewhat confused by this discrepancy.

We do not have independent access to the transverse susceptibility – the only access we have to it is through our measurement. Thus, the dashed curves are estimates based on the longitudinal magnetization and longitudinal susceptibility (i.e. the field derivative of the longitudinal magnetization) alone. The difference between the measured and the dashed, “predicted” curves suggests that transverse susceptibility is non zero at high ($B > 10$ T) fields along the c-axis and nearby in the phase diagram.

2) In Fig. 2d of the main text (as well as in Supplementary Fig. 4b), a jump is observed in the magnetotropic susceptibility k_a , for large angles ($\theta > 59^\circ$) and at approximately 40–50 T. This jump is used to identify the metamagnetic transition. However, it is not entirely clear to me whether this jump originates from the first or the second term in Eq. (1).

The case at $\theta = 59^\circ$ is near the QCEP, where the jump/dip is most pronounced. Since the longitudinal component of magnetization is not expected to exhibit a jump at the QCEP — given that it constitutes a continuous quantum

phase transition — I suspect that the jump in k_a arises primarily from transverse fluctuations. Is this interpretation correct? In any case, additional discussion on this point seems warranted.

This interpretation is correct – at the QCEP, the jump in k_a has to come from the second term. To say anything further, we should first clearly state what we mean by fluctuations: our magnetotropic measurements are quasistatic (the resonance frequency is of order 40 kHz), and thus what we mean by “fluctuations” is a large increase in the observed susceptibility (the magnetotropic coefficient). By the fluctuation-dissipation theorem, this implies increased fluctuations. For example, we take the large softening in k when magnetic field is applied parallel to c to be an indication of strong transverse fluctuations.

Referring specifically to the jump: the jump in k (or any susceptibility) at a second order phase transition—including the QCEP—is fully expected within mean field theory (see e.g. <https://www.tandfonline.com/doi/abs/10.1080/00018737300101379>). Thus while in equation 1 we can attribute the jump to the second term (to the transverse susceptibility), it is not directly from the fluctuations *per se*.

Furthermore, could authors clarify how the transverse susceptibility χ_{\perp} presented in the contour plot Fig. 4 was obtained?

In Figure 4, the magnetotropic susceptibility k multiplied by $-\mu_0/B^2$ is plotted as a function of field angle (as described in the figure caption). In the limit that the transverse susceptibility is much larger than the longitudinal susceptibility, $-\mu_0 k/B^2$ is equal to χ_{\perp} . Figure 3 illustrates the degree to which this approximation is valid in UTe₂. The solid curve χ_{bb} is χ_{\perp} for this field direction (B||c), and is calculated using the known longitudinal magnetization and equation 1 (i.e., we subtract off the first term). The dashed curve, $-\mu_0 k/B^2$, is a very good approximation to χ_{bb} , at least when χ_{\perp} dominates. Note that we are only able to perform this exact extraction of χ_{\perp} for directions where M vs H has been measured. However, given the apparent large size of the transverse susceptibility across the entire angle range, we use $-\mu_0 k/B^2$ as a qualitatively good proxy for χ_{\perp} everywhere in Figure 4. **We added more explanation in the caption of Figure 3 to help clarify.**

3)A particularly intriguing finding is the QCEP and the strong FM fluctuations detected via k_n in its vicinity. This discovery establishes a quantum analogue of the critical endpoint (CEP) and supercritical states in a classical liquid-gas phase diagram. Relatedly, in the ground-state phase diagram of Ising model under longitudinal and transverse fields, a first-order transition line terminates at a QCEP, giving way to a quantum supercritical regime with strong FM fluctuations.

Furthermore, the measured deviation of B_{\max} from B^* may also arise from finite-temperature effects, as the experiments were conducted at 4 K (well above the SC transition). The influence of a QCEP at finite temperature has also been explored within the quantum Ising model, characterized by large susceptibility (corresponding to the transverse susceptibility here) amongst other thermodynamics (see arXiv:2410.22236).

First, thank you for bringing the analogy to the liquid-gas phase diagram to our attention. **We have now referenced the article mentioned above in the discussion.**

In Figure 4, we decided to remove the label B_{\max} from the phase diagram. B_{\max} was meant to signal the region where the observed transverse fluctuations were maximum, but we think that it incorrectly signaled a critical endpoint, especially within the context of this community.

minor points:

i)Caption of Fig. 1, “ $k_b(B||b) = B (M_b - \dots)$ ” seems a typo, should it be “ $k_a(B||b)$ ”?

Thank you, this has now been corrected.

ii)In Figure 1a, the metamagnetic transition forms a continuous boundary. This appears to contrast with Figure 4, where a QCEP terminates the first-order transition line.

The phase diagram in Figure 1 is reconstructed from the data in Lewin *et. al.* [Science 389;6759 pp 512-515 (2025)] and Ran *et. al.* [Nat. Phys.15:125-1254, (2019)]. In Figure 4, the phase diagram was previously constructed based on our data, where the small angle misalignment was not included. The angle misalignment (now described in detail in the “crystal axes and alignment” section of the SI) resulted in raising the metamagnetic transition and lowering the critical endpoint, in agreement with our observations. **We have changed Figure 4 to accurately reflect that the field orientations correspond to measurements done in a plane slightly misaligned from the bc-plane.**

iii)References [7] and [14] (currently cited as arXiv preprints) have since been published. The published versions can be cited instead.

This has been fixed. Thank you.

Thank you for the close reading of our manuscript.

We sincerely appreciate each of the reviewers' time for their insightful and detailed comments. We feel that they have greatly improved the clarity and accuracy of our manuscript.

REVIEWERS' COMMENTS

Reviewer #1 (Remarks to the Author):

In my previous report, I requested further measurements to clarify some problems (similar concerns were also raised by other referees). The authors failed to perform these measurements and instead provided analyses of the sample misalignment based on the comparison with some previously published results. I can understand that such measurements are time consuming. On the other hand, their arguments seem reasonable. Therefore, I'm fine if the editor is going to publish this manuscript in NC.

BTW, the sequence of the references should be sorted out.

We updated the references to appear in accordance with the order of citations in the manuscript.

Reviewer #2 (Remarks to the Author):

The authors have performed an admirable job in patiently responding to each of the referees' questions. The efforts to quantify and fairly present the angular misalignments are very robust. I have a few small suggestions listed below, which could help further strengthen the paper's impact:

Firstly, I spotted a small typo in the supplement on page 3: "according the angle" should be "according to the angle".

The field of UTe₂ still seems to be moving quickly, and there are a number of recent results that the (still rather short) discussion section might consider expanding on. Notably, in Lewin et al PRB 110, 184520 (2024) and Wu et al arXiv:2503.11362 anomalous features in the UTe₂ high field phase landscape have been observed in conductivity measurements. These could possibly be related to the present work(?)

The introduction briefly mentions the disorder sensitivity (or lack thereof) in the Frank et al 'orphan' paper. However, a fascinating recent study looking at thorium-substitution (Moir et al, PNAS e2521261122 2025) has shown that SC1 can be preserved while SC3 is suppressed! This points to a leading role for dimer?

Also, clear signatures of non-Fermi liquid behavior have been resolved in arXiv:2505.12131 and arXiv.2505.09351. These seem to correspond well to where the B* star is located in Fig 4 of the present study. This seems like very nice corroboration between 3 independent groups (including this work).

In summary, I find the additions to the supplement to strongly enhance the credibility of the paper, which provides a very interesting new perspective on the high magnetic field properties of the enigmatic superconductor UTe₂. I list the above suggestions merely for the authors' consideration, and strongly support the paper's publication in Nat Comm.

We fixed the typos. Thank you for the comments related to recent studies. Where appropriate, we have tried to incorporate these references. For example, in the discussion we now point out that the observed fluctuations could be related to the T² and T-linear behavior near SC3 (preprints mentioned above). However, in the case of the work by Lewin et al PRB 110, 184520 (2024) and Wu et al arXiv:2503.11362, they observe a magnetoresistive hump in a different measurement plane than we consider in our work so the connection is not immediately clear to us.

Reviewer #3 (Remarks to the Author):

After reading the revised manuscript and the authors' detailed response to the reviewers' comments, I find that the authors have addressed the points carefully, and the overall quality of the work has been improved. While it is still not entirely clear to me (i) the origin of the jump in Fig. 2d, and (ii) why the superconductivity region appears to repel the strong transverse ferromagnetic fluctuations, I believe the experimental results are substantial and deserve publication in their current form.

I have only one minor point for the authors' consideration. Although the description of B_{max} has been removed, the remaining discrepancy might still originate from quantum fluctuation effects. The phase boundary in Fig. 4 is determined at a finite temperature. If one imagines lowering the temperature further, this boundary should still

take the form of a first-order transition line that terminates at a critical endpoint (CEP). The zero-temperature limit of this CEP would constitute a quantum critical endpoint (QCEP). This QCEP is expected to be located at a field distinct from B^* and could strongly influence low-temperature magnetothermodynamics — including the magnetotropic response measured here.

We agree with your description of the phase boundary. It will be very interesting to perform these measurements at lower temperatures. We plan to do this while considering different experimental geometries (i.e. different measurement planes).